# Advanced Computational Methods for Agriculture Machinery Movement Optimization with Applications in Sugarcane Production

**Martin Filip [1,]\*** , **Tomas Zoubek [1]** , **Roman Bumbalek [1]** , **Pavel Cerny [2]** , **Carlos E. Batista [3]** , **Pavel Olsan [1]** , **Petr Bartos [1,2]** , **Pavel Kriz [1,2]** , **Maohua Xiao [4]** , **Antonin Dolan [1]** and **Pavol Findura [1]**

1  Faculty of Agriculture, University of South Bohemia, Studentska 1668,
   370 05 Ceske Budejovice, Czech Republic; tomzoub@gmail.com (T.Z.); bumbalekr@zf.jcu.cz (R.B.);
   pavel.olsan@gmail.com (P.O.); bartos@zf.jcu.cz (P.B.); kriz@pf.jcu.cz (P.K.); dolan@zf.jcu.cz (A.D.);
   pavol.findura@uniag.sk (P.F.)
2  Faculty of Education, University of South Bohemia, Jeronymova 10,
   371 15 Ceske Budejovice, Czech Republic; pcerny@pf.jcu.cz
3  Faculty of Engineering of Ilha Solteira (FEIS/UNESP), São Paulo State University, Passeio Monção 830,
   15385-000 Ilha Solteira, Brazil; visone@reitoria.unesp.br
4  College of Engineering, Nanjing Agriculture University, Nanjing 210031, China; xiaomaohua@n.jau.edu.cn
\*  Correspondence: filipm07@zf.jcu.cz; Tel.: +420-387-772-929

**Abstract:** This paper considers the evolution of processes applied in agriculture for field operations developed from non-organized handmade activities into very specialized and organized production processes. A set of new approaches based on the application of metaheuristic optimization methods and smart automatization known as Agriculture 4.0 has enabled a rapid increase in in-field operations' productivity and offered unprecedented economic benefits. The aim of this paper is to review modern approaches to agriculture machinery movement optimization with applications in sugarcane production. Approaches based on algorithms for the division of spatial configuration, route planning or path planning, as well as approaches using cost parameters, e.g., energy, fuel and time consumption, are presented. The combination of algorithmic and economic methodologies including evaluation of the savings and investments and their cost/benefit relation is discussed.

**Keywords:** optimization; agricultural machinery; metaheuristic algorithm; precision agriculture; route planning

## 1. Introduction

Physical optimization has been key to increasing the productivity and efficiency of agricultural machinery for decades. This is related to the savings resulting from the improved mechanical functionality of the machines. Currently, there are environmental and biological factors that prevent further developments in the field of mechanical optimization. In particular, it is a matter of limiting the size and weight of the machines, where soil compaction machines can be mentioned as an example [1].

Analysis of the planned operation, operations optimization focusing on the route optimization and capacity dimensioning, as well as operation planning (including resource allocation, scheduling, analysis of the time required to carry out the operation and mission planning) are integral parts of modern agricultural operations management which supplement traditional agricultural operations planning methods [2].

Considering the current development of automation systems using communication and information technologies, it is possible that human sensory and mental inputs could be replaced by such systems. The use of these automation systems provides a number of emerging benefits, including increased repeatability associated with increased work performance and increased capacity. In addition, labor costs and the use of material inputs (e.g., agrochemicals and fertilizers) decrease. Moreover, the flexibility of the production system increases due to the easier adoption of new production practices. Automation systems also provide better control of processes, leading to the increased quality of products [3].

Sustainability in production systems entails increased demands for decision-making during operational management considering environmental impacts [2].

Reduction of non-productive times is key to reducing production costs. These non-productive times reduce overall efficiency and include agricultural machinery operations in the field for which they were not primarily acquired to perform, such as loading and offloading agricultural inputs and turning while moving in the field [4].

In agricultural machinery management and optimization, route planning issues in field operations can be considered fundamental [3]. Agricultural route planning must include fleet parameters, managed tasks and optimized operations. If we consider the vehicle fleet, the type and number of vehicles, the mode of operation and consumption are important. Important managed tasks include tillage, sowing, watering, fertilizer application and harvesting. Optimized operations consider, in particular, the time and cost required [5–8]. Approximately 30–35% of production costs are related to the harvest. On average, a sugar cane harvester spends only around 36% of its time working efficiently [9]. It is important to minimize the number of maneuvers that makes the machine's performance more efficient [10]. Aires also reported that it is possible to increase the operational yield of the harvest by the systematization of plots [11]. Salvi et al. highlight that the mechanized system management before and during the project execution becomes relevant for cost reduction. In this study, the authors found that a 5% increase in operating efficiency reduces the cost of mechanized cutting by 8% [12]. Optimal use of the machinery will significantly reduce production costs.

This review aims to discuss modern computational approaches which take place in agriculture machinery movement optimization with applications in sugarcane production. In the third and fourth section, we review the agricultural routing problem (ARP) and the most common metaheuristic algorithms for agricultural applications, respectively. In the fifth section, we demonstrate the benefits of the application of these approaches in sugarcane production, which is one of the most important crops in Latin America.

## 2. Materials and Methods

Web of Science (WoS) and ScienceDirect (SD) databases for bibliometric analysis of scientific publications were mainly used. These databases contain original research and review articles, book chapters and other publications with the highest level of quality. For this reason, WoS and SD were used as the main sources of information in this study. In addition, publications obtained from the mentioned databases were supplemented by articles found in Springer Link (indexed by Scopus) and Wiley Online Library.

The scope of research included publications in materials from the Elsevier, Cambridge University Press, IGY Global and ASABE publishers. Moreover, publications from Brazilian authors were included in the review in view of the particular importance of sugarcane production in Brazilian agriculture. All publications in the WoS and SD, searched by key words "route planning", "path planning", "sugarcane production", "heuristic optimization methods" and "heuristic algorithms", published from 2010 to the present were analyzed. Exceptionally, several older publications were added for their relevance.

A qualitative bibliometric method based on the analysis of publications by the authors themselves, partially supported by WoS sophisticated tools and the extraction of bibliometric data for processing in

a spreadsheet software, was applied in the work. The used methodological approach included the following stages:

1.  Identifying publications in scientific databases by key words: "route planning", "path planning" and "sugarcane production".
2.  Analysis of the results and selection of relevant publications of journals focused on agriculture and technology using the "Analyze Results" tool (WoS).
3.  Downloading all selected relevant publications in the analyzed period and extracting their bibliometric data (authors, title, year of issue, key words, additional key words, publishing house) using the "Export to Excel" (WoS) and "Extract" (SD) tools.
4.  Processing of bibliometric data using spreadsheet software, MS Excel 2019 (sorting according to required criteria, identification of articles from the same authors, key words analysis for further search).
5.  Detailed qualitative analysis of the content of selected publications in terms of:

    a.  investigated problem/topic,
    b.  area of application,
    c.  used type of method/algorithm,
    d.  achieved results and their relevance to the solution of the investigated problem.

## 3. Agricultural Routing Problem (ARP)

Field efficiency defined as a rate of agricultural machine performance during field operations is calculated from the ratio between the real field productivity and the theoretical maximum productivity of the machine [13]. Field efficiency depends on several factors. Thus, it is not constant for a particular agricultural machine. Among others, these factors include the size and shape of the field, crop yield and moisture. However, the most important factor is the pattern of field operation, which significantly affects time lost during non-working travel in the field [14].

Generally, coverage path planning (CPP) deals with issues related to route planning in order to pass through all points of the defined area and avoid all obstacles [15]. Coverage path planning specifically applied in the agricultural area is called agricultural routing problem (ARP). The ARP problem deals with issues related to the minimization of the traveled distance by agricultural machines without restrictions on route coverage [16]. The solution of the ARP problem usually aims to minimize input costs, use more machines and optimize the required time, while it is complicated by obstacles, the limited production capacity of the machine or the requirement for use in multiple fields [17].

The recent study conducted by Bochtis and Vougioukas dealt with the agricultural routing problem and was focused on the calculation of the decomposition of a complex-geometry field into sub-fields as well as on the determination of the optimal driving direction in each field [18].

Maneuvers of the machines in agriculture field operations can be classified according to the shape of their trajectory. The three most common types of maneuver ($\Omega$-turn, U-turn and T-turn) are shown in Figure 1 [18,19]. The behavior of the maneuver and required space are affected by the angle between the field border and the working direction [19,20]. To make $\Omega$-turn, more space compensated by the higher maneuver speed is needed, while the machinery returns to an adjacent row. On the other hand, this maneuver requires wider headlands and more area taken out of the production [21].

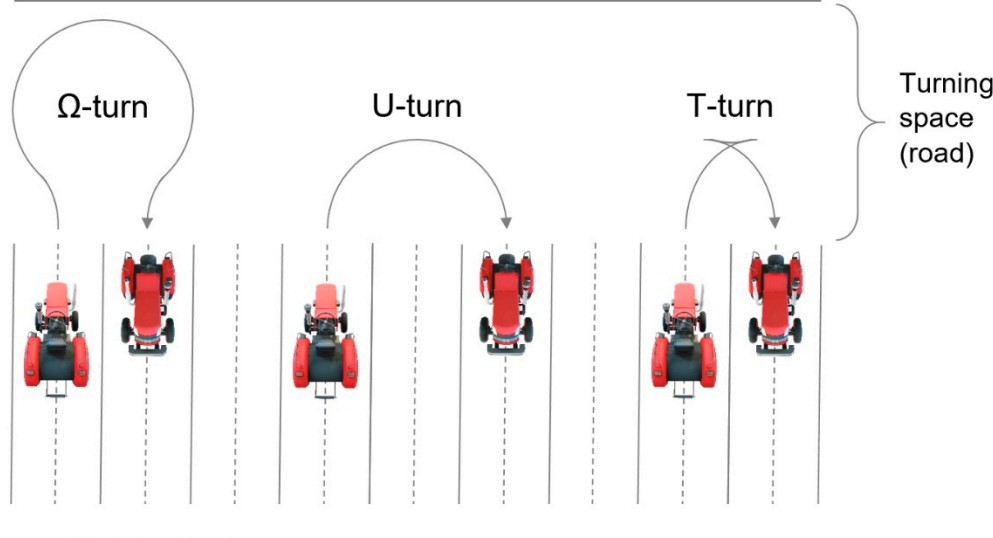

**Figure 1.** Types of common maneuvers in field operations.

U-turns require less space and are the fastest, but at least one of the following two conditions needs to be met. Either the implement width must be larger than the diameter of the turning circle or the rows must be skipped [18].

It can be seen that T-turns require low maneuvering space to reach the adjacent track in the case of sugarcane operations [22]. If necessary, the guidance systems ensuring parallelism and correct skipping distance from the previous row can be used to shift from T-turns to skipping rows.

## 4. Metaheuristic Algorithms for Agricultural Applications

A metaheuristics is a combination of heuristic methods that aims to effectively promote the exploration of the search space. Thus, it is possible to overcome the local search trap in a complex space. It can also be understood as a general heuristic method that is developed to guide a specific heuristic (a larger strategy for developing smaller or heuristic methods). A metaheuristic is inspired by several topics that can be highlighted, such as analogies with physical, chemical, biological, semantic and social phenomena. They can be classified according to the neighborhood structure employed or the strategy of obtaining the solution [23].

Different authors classify metaheuristic methods in different ways [24,25]. Darwish et al., Dhiman and Kumar and Khishe and Mosavi use four basic groups: evolutionary computing, swarm intelligence, human- and society-inspired and physics-based [26–28]. Aladeemy et al. divided metaheuristic methods very similarly, but they merged the evolutionary computing, swarm intelligence and physics-based groups into one large group called nature-inspired algorithms [29]. On the other hand, Elshaer and Awad and De Leon-Aldaco et al. divided metaheuristic methods into only two groups, which they called single and population-based [30,31]. Francik et al. also included groups called evolutionary computation and swarm intelligence in their review of current research trends in renewable energy source systems [32]. Hegazy et al. divided metaheuristic methods into three groups: evolutionary computation, swarm intelligence and physics-based [33].

Based on a detailed analysis of the available literature dealing with metaheuristic methods and their classification, a comprehensive overview of important metaheuristic methods is provided in Figure 2. The most important metaheuristic methods used in precision agriculture with regard to sugar cane production are highlighted in color and described in detail below. Stated metaheuristic methods were divided into four groups: evolutionary computing, swarm intelligence, physics-based algorithms and human-inspired algorithms.

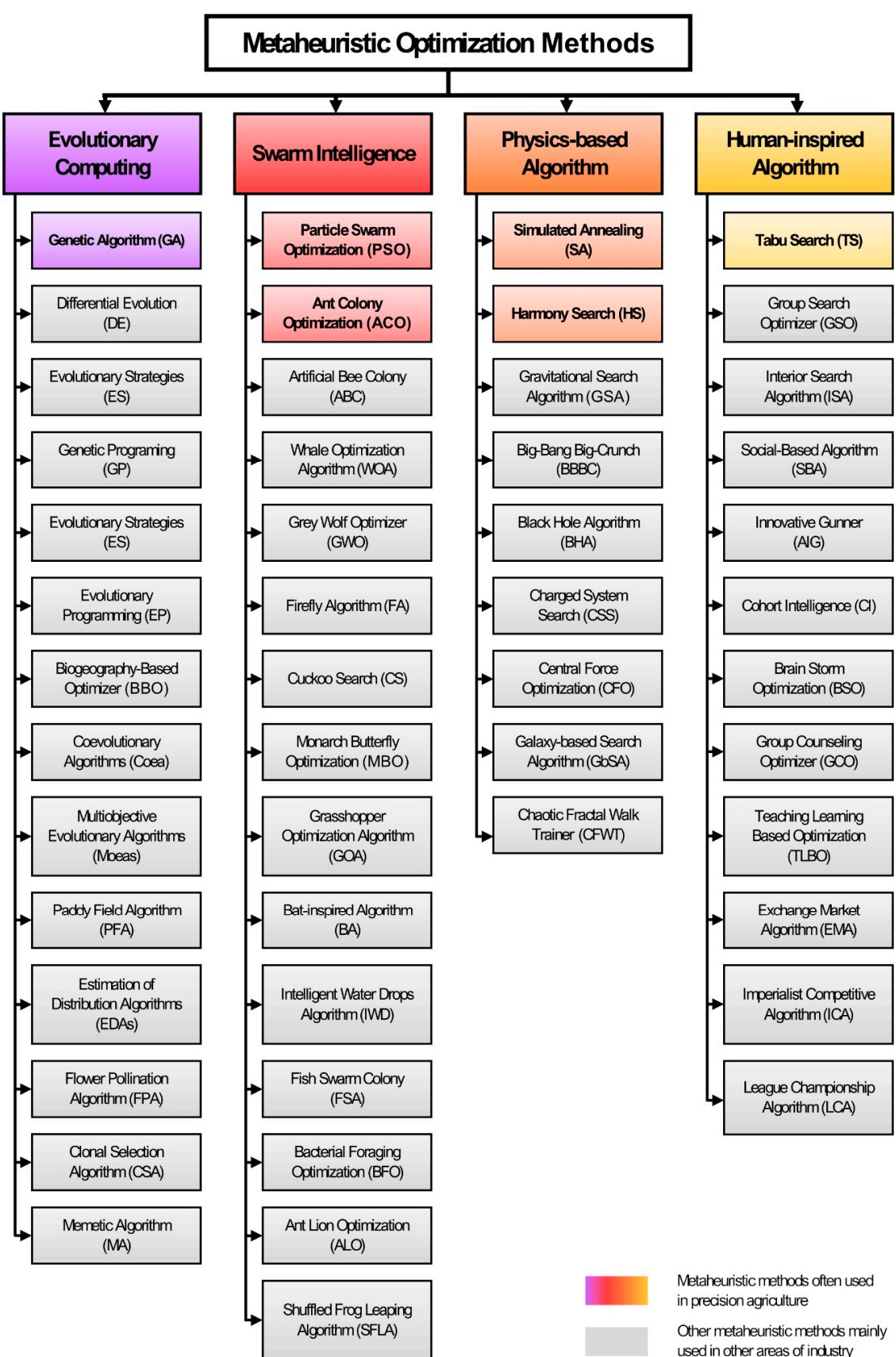

**Figure 2.** Classification of metaheuristic optimization methods.

The evolutionary computing group includes genetic algorithms (GA), differential evolution (DE), evolutionary strategies (ES), evolutionary programming (EP), biogeography-based optimizer (BBO), coevolutionary algorithms (Coea), multiobjective evolutionary algorithms (Moeas), paddy field algorithm (PFA), estimation of distribution algorithms (Edas), flower pollination algorithm (FPA), clonal selection algorithm (CSA) and memetic algorithm (MA).

The swarm intelligence group includes particle swarm optimization (PSO), ant colony optimization (ACO), artificial bee colony (ABC), whale optimization algorithm (WOA), gray wolf optimizer (GWO), firefly algorithm (FA), cuckoo search (CS), monarch butterfly optimization (MBO), grasshopper optimization algorithm (GOA), bat-inspired algorithm (BA), intelligent water drops algorithm (IWD), fish swarm colony (FSA), bacterial foraging optimization (BFO), ant lion optimization (ALO) and shuffled frog leaping algorithm (SFLA).

The group of physics-based algorithms includes simulated annealing (SA), harmony search (HS), gravitational search algorithm (GSA), big-bang big-crunch (BBBC), black hole algorithm (BHA), charged system search (CSS), central force optimization (CFO), galaxy-based search algorithm (GbSA) and chaotic fractal walk trainer (CFWT).

The human-inspired algorithm group includes tabu search (TS), group search optimizer (GSO), interior search algorithm (ISA), social-based algorithm (SBA), innovative gunner (AIG), cohort intelligence (CI), brain storm optimization (BSO), group counseling optimizer (GCO), teaching learning-based optimization (TLBO), exchange market algorithm (EMA), imperialist competitive algorithm (ICA) and league championship algorithm (LCA).

As can be seen in Table 1, several metaheuristic methods have been proposed to solve the CPP and agricultural field logistic problems [18]. The most common algorithms applied to the problem are genetic algorithm, ant colony algorithm, simulated annealing, harmony search, particle swarm optimization and tabu search.

**Table 1.** Overview of the most common metaheuristic methods for the solution of the coverage path planning (CPP) and agricultural field logistic problems.

| Method Name Abbreviation | Method Name |
|:---:|:---:|
| GA | Genetic Algorithm |
| ACO | Ant Colony Algorithm |
| SA | Simulated Annealing |
| HS | Harmony Search |
| PSO | Particle Swarm Optimization |
| TS | Tabu Search |

*4.1. Genetic Algorithm (GA)*

Genetic algorithms are used among heuristic methods. Their basic idea is inspired by biological evolution, so they work on the principle of the mechanism of natural selection and genetics [34,35]. The development of genetic algorithms began in the 1960s and is attributed to J. Holland [36]. Mohammed et al. proposed that genetic algorithms can be effectively used to find the optimal solution if the space of all solutions is too large and linear programming cannot find a theoretical solution on time and for solving problems with multiple constraints [37]. The main advantages include the ability to optimize problems with many solutions, the use of parallel data selection, which eliminates deadlocks in a locally optimal solution, and good intelligibility of the algorithm [38]. Lamini et al. state that the genetic algorithm contains five basic parts, namely the selection of values of variable parameters of the genetic algorithm, such as population size, crossing probability, mutation probability and criterion for stopping the algorithm. Other parts are composed by suitable data coding, a method of generating the initial population, a fitness function used to evaluate the quality of each potential solution and genetic operators that modify the genetic composition of the parental chromosomes to create new chromosomes (offspring) [39]. At the beginning, the data are encoded into chromosomes and a random population is

generated [36]. Then, each chromosome is evaluated using the fitness function, which calculates the quality of the chromosome, also called fitness, which is an important criterion for the next step, which is selection. A new generation of chromosomes (offspring) will be created from selected chromosomes (parents) using other genetic operators, which are crossing and mutation. For this, the quality is calculated, and then their inclusion creates a new generation of the population [40]. This process is repeated until there is a quality improvement that meets the optimization criterion [41].

Mohammed et al. used improved genetic algorithm to solve the vehicle routing problem strongly related to agricultural routing problem. The main goal of their study was to reduce the consuming time as well as the distance of vehicle paths and, consequently, the transportation costs. They concluded that the improved genetic algorithm can be considered universal in relation to a number of real-life applications. It is active in solving numerous different issues including agricultural routing problems [37]. Kurnia et al. also used the genetic algorithm to solve the vehicle routing problem and pointed out that GA is able to provide optimal results by searching a solution with a satisfactorily short total distance [35]. Wieczorek & Ignaciuk used modified Genetic Algortihm to optimize processes of reflow in the network channels [42], while Gracia et al. investigated the use of the hybrid genetic algorithm to solve the bale collecting problem. They used GA to optimize resources such as minimizing non-productive time, fuel consumption or traveled distance. The results obtained due to the use of hybrid genetic algorithm showed an average improvement of savings by 17% from those obtained by previously tested nearest neighbor heuristics [43]. Lamini et al. used a genetic algorithm-based approach for autonomous mobile robot path planning potentially usable in precision agriculture. An improved crossover operator using the genetic algorithm in a static environment was suggested to solve path planning problems. The results of their simulation showed that using GA with the improved crossover operators helps to find solutions with very promising parameters compared to other methods. Suggested approach and its parameters were based on reducing the number of turns to reach the goal, leading to optimization of the energy consumed by the mobile robot [39]. Sales et al. integrated Monte Carlo simulation with the genetic algorithm in order to solve the field layout design problem. In this paper, three case studies are presented. GA supplemented by Monte Carlo simulation provided many novel possible approaches in the studied production system. The output of such a simulation also included a large amount of information to support the decision-making process. Based on the presented results, Sales et al. discussed possible perspectives of the use of the probabilistic approach with potential use in the field of development engineering [44]. Neungmatcha et al. successfully used the comprehensive decision support system with a geographical information system based on the adaptive genetic algorithm (AGA) in the sugarcane supply system to reduce costs and to design efficient management. They stated that AGA provided better results in comparison with current practices. The point of their study was to solve problems associated with the planning of the transportation and allocation from fields to the loading stations in sugarcane production systems [45]. Tong et al. also investigated a modified genetic algorithm known as greedy genetic algorithm (GGA). They used GGA for path optimization of seedling low-density transplanting. The authors compared conventional genetic algorithm and greedy genetic algorithm and concluded that the average operation time of GGA was 71% lower compared to GA [46]. Gracia et al. published a study that dealt with collecting sequences of residual biomass. They used a hybrid genetic algorithm and local search heuristics significantly reducing the total distance compared with other common assignment rules. As the authors stated, the reduction of the total distance depends on the loading capacity constraint and on the type of crop. Total distance refers to the sum of working and non-working distance (composed of non-working in-field distance and non-working out-field distance). It can be reduced by up to 20% [47].

### 4.2. Ant Colony Algorithm (ACO)

Ant colony algorithm is an optimization method that comes from the behavior of ants searching for food around their nest. Movement of ants is random, but ants go back by the same path when

they find food and they leave pheromones on their paths, which is attractive for other ants that follow the pheromones path [48]. This communication between ants is called indirect communication and it is directed by the intensity of the pheromones path, which is affected by the number of passing ants and frequency of their passes because pheromones evaporate over time [49]. The basic principle is that, at the beginning, the pheromone concentration on possible paths is zero and the probability of path selection is the same. The ants move at the same speed, but those who choose a shorter path will reach the food sooner and thus begin to release pheromones earlier on the way back. The shorter path is thus covered by pheromones earlier and it begins to attract more ants. The probability of selection is proportional to the intensity of pheromones on a path in the current time. The intensity of the pheromones increases and thus the probability of selection for the shortest path increases too after several iterations [50]. The main parameters of the algorithm include pheromone concentration, maximum iteration time, population size, number of variables, importance degree of pheromone and inspiring factor [51].

Bakhtiari et al. were engaged with the optimization of harvesting process. They used the ant colony method for searching the ideal paths for the combine harvester, which must take the harvested crop out of the field (variant of harvesting process without haulers on the field). The results of the work showed a reduction of non-working routes by 19.3–42.1% and savings of non-working distance of 18–43.8% [52]. Sethanan and Neungmatcha introduced a competitive algorithm for sugarcane field work based on particle swarm optimization [53]. Feng dealt with the logistics model potentially applicable in precision agriculture based on the ant colony algorithm as well as artificial intelligence methods. He summed up the benefits of the ant colony algorithm (ACO) and its universality including the absence of a need for the initial route. Moreover, Feng also concluded that ACO does not depend on the choice of sub-initial route and it does not need to be adjusted manually in the search process [50]. Ant colony algorithm can also be used in the case of electric vehicles, whose importance in agriculture will increase significantly in the future [54]. Joo and Lim introduced an effective energy efficient routing method using ACO to maximize the energy efficiency [55]. The efficiency of the transportation of products to specific places and improvement of handling capacity was investigated by Mutar et al. They used ant colony algorithm and concluded that the mathematical results showed that the proposed method seemed very promising and provided better results than other algorithms (e.g., SA) [56]. Ant colony algorithm was also used for improving the energy efficiency of road vehicles. Donati et al. used the ant colony algorithm to increase the energy efficiency of road vehicles. The results were used as a basis for assessing the costs of different levels of carbon dioxide standards for vehicles in Europe [57]. Ali et al. improved the efficiency of the optimization by the ant colony algorithm using additional assistance of A* multi-directional algorithm. Their work dealt with path planning problems in static and dynamic constraints environment and its application on mobile robots in a real-world environment. Similar studies using ACO optimization were also published by Wang and Pu et al. Wang focused on unmanned wheeled robots, while Pu et al. focused on optimization of three-dimensional paths in robotics [58–60]. In addition to its use for solution of vehicle routing problems and path planning, ACO can also be used to identify spectral signatures of fruit and vegetables, as reported in [61].

### 4.3. Simulated Annealing (SA)

Simulated annealing is a stochastic algorithm proposed by Kirkpartick, Gelett and Vechchi in 1983 [62], which is applied to find the optimal solution of the objective function as the global extreme of function with local minimums [63]. SA is inspired by the physical annealing process of metals [64]: when the process begins at a high temperature, physical elements of the material obtain enough energy to escape from their start positions (local minimums), and gradual cooling leads to thermal equilibrium. Initial temperature, cooling rate and termination policy are primary features of SA [65]. The inner part of the algorithm is composed by the Metropolis procedure, which generates new solutions by the small probabilistic movement of the existing solution; the best solution to be achieved is determined by iterative steps [63,64].

Cerdeira-Pena et al. introduced, implemented and tested two different heuristic algorithms based on tabu search and simulated annealing philosophy, respectively. They applied this model to a real-world problem related to an agricultural cooperative that needed to optimize the routes of several harvesters. Obtained results showed that the cooperative successfully adopted SA heuristics as part of an agricultural management operation tool [66]. Grabusts et al. applied SA heuristics for optimal route detection between objects using GPS location. They developed the software for searching and optimizing of the shortest route between different objects [64]. Simulated annealing algorithm was also used in optimization thinning schedule of a forest stand. It was enhanced by Moriguchi in terms of its calculation speed and reliability [67]. Small unmanned aerial vehicles, becoming very popular in precision agriculture, also require sophisticated path planning. This can be carried out by the simulated annealing algorithm, as reported in [68]. Simulated annealing was also successfully applied for agricultural monitoring. For example, Leitold et al. proposed a simulated annealing-based methodology to assign additional sensors to dynamical monitoring systems [69]. A comparison of the two different developed solutions is shown in Figure 3. The approach proposed by Conesa-Muñoz et al. showed a total traveled distance of 334.439 m and a distance traveled on the headlands of 94.439 m [70]. These results can be compared with the results reported in [18]. Bochtis and Vougioukas proposed an approach based on simulated annealing used to solve the route planning problem for herbicide applications. Performed experiments were carried out in four fields. Obtained total traveled distance was 335.767 m and the distance traveled on headlands was 95.767 m [18]. By comparing both results, it can be stated that the differences between calculated distances are less than 1.4%.

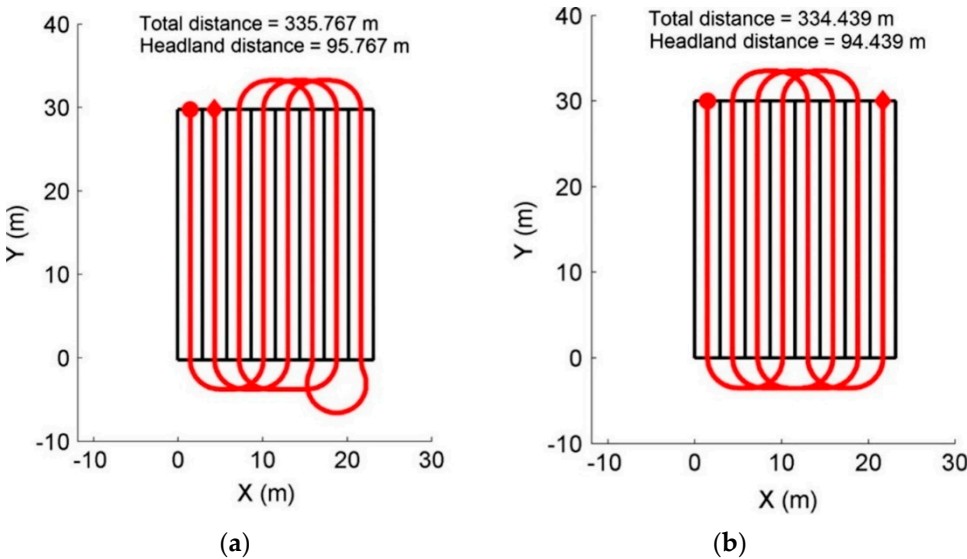

**Figure 3.** (**a**) Solution proposed by Bochtis and Vougioukas in 2008 [18]; (**b**) solution proposed by Conesa-Muñoz et al. in 2016 [70].

### 4.4. Harmony Search (HS)

This optimization method is inspired by real-world processes like many other metaheuristic optimization methods, concretely in the process of musical improvisation in search of a melody [71]. At the beginning, harmony is generated like a composition of random notes and it is stored in memory. Generated notes are tuned to new harmony by a small shifting of their tones [72]. New harmony (group of variables) is created by each iteration of the algorithm and mutations of the variables are again made by their small change or complete replacement. Harmony selection and memory update are executed at the end of each iteration. The main advantages of the harmony search are the possibility to consider every member of the population when generating new solutions, preserving precision

when a continuous decision variable is participating in improvisation step and random initialization of decision variables [73].

Probably the most common use of the harmony search algorithm in agriculture is related to the agricultural routing planning problem. For example, Utamima et al. used the HS algorithm for agricultural routing planning in field logistics [74]. Like other heuristics, the harmony search algorithm was also used for optimization of aerial vehicles in precision agriculture management and path planning in robotics, respectively [72,73]. Researchers are making great efforts to refine existing HS algorithms. For example, Liu et al. introduced a method using the HS algorithm based on natural coding for a vehicle routing problem. This method adopted natural number coding with high efficiency and provided better results compared to other tested types of HS algorithm [71]. Valente et al. developed an approach using the harmony search algorithm for aerial coverage optimization. Barrientos et al. achieved a higher degree of optimization of agricultural machinery routes using the harmony search algorithm. The main feature of the approach is to reduce the number of turns of the trajectories while holding the start and finish positions [75]. The disadvantage of this approach is the longer computing time [72].

### 4.5. Particle Swarm Optimization (PSO)

A particle swarm optimization algorithm was first proposed by Kennedy and Eberhart [76]. This algorithm is based on searching the D-dimensional space of a swarm of particles, which are looking for a globally optimal solution to a problem in this space. The speed of a particle is a recalculation based on knowledge of its fitness so far or on the experience of other members of the swarm [77]. Xun Li et al. describe particle swarm optimization as a metaheuristic method commonly used in applications as well as in route planning research. This method is inspired by the behavior of a flock of birds and is characterized by a high crawl rate. The main advantages are easy implementation of the method and fast convergence to the optimum for a large range of functions. The disadvantage of the particle swarm optimization method is the insufficient convergence speed for robot route planning, so Xun li et al. proposed an improved PSO integration scheme [78].

Particle swarm optimization method can be used in smart farming for capacity planning of agricultural machinery maintenance service [79] as well as for unmanned aerial vehicles. Mukherjee et al. applied swarm processing in processing-intensive tasks such as visual identification of farmlands, crop health monitoring and crop growth tracking [80]. Probably the most important area with an overlap into smart and precision agriculture where PSO is used is robotics. Das and Jena and Li et al. used the PSO algorithm for path planning of multi-robots and mobile robots, respectively [77,78,81].

### 4.6. Tabu Search (TS)

Tabu search algorithm was first proposed by Glover [82]. The tabu search was employed for the optimization of logistics problems [83]. The tabu search method is based on the local search method. The tabu search algorithm creates a tabu list when it searches. The tabu list contains solutions to which the tabu search should not return because it has already been processed [84]. The tabu list has a fixed length *n*, which lists the last *n* changes (variable, value) [85]. The length of the tabu list is an input parameter of the algorithm [84]. When selecting a value for a variable, it is determined whether this pair is no longer recorded in the tabu list [86]. The status of prohibited changes depending on time and circumstances is based on evolving memory. In algorithms implementing the tabu list, there is a function that allows the selection of a solution, even if the solution is in the tabu list. This function is called the aspiration criterion [85]. Choosing a prohibited exchange can occur when a better solution has not yet been obtained [87]. The tabu search, thanks to the tabu list, prevents trapping at a local minimum of the optimization function [83]. The tabu search achieves a better solution in several local search spaces [88]. However, comparing a solution with the tabu list takes more time [89].

Like other algorithms, probably the most common use of the tabu search algorithm in agriculture is related to the agricultural routing planning problem. For example, Utamima et al. used TS algorithm together with HS algorithm for agricultural routing planning in field logistics [74]. Sethanan et al. proposed a mathematical model to optimize mechanized sugarcane harvesting with the objective of maximizing the yield of sugarcane percentage. To solve the model, two heuristic algorithms were used. The first algorithm aimed at scheduling the mechanized harvest, while the second algorithm aimed at optimizing the solution of this algorithm using a tabu search. The results showed a 16.38% improvement in sugar production [90]. He et al. created an operational model of the combine harvester using the hybrid tabu search method. The result showed the possibility of reducing the harvest time by 10% on fragmented agricultural land [91]. Hybrid algorithm using both adaptive large neighborhood search and tabu search was used to reduce non-working distance in field logistics for heterogenous harvesters [88]. He et al. also proposed a two-stage algorithm constructed by a tabu search to optimize the non-working distance for heterogenous vehicles [85]. Seyyedhasani et al. introduced routing algorithm selection for field coverage planning using two routing algorithms, Clarke–Wright and tabu search algorithm. They concluded that TS algorithm provided better result compared to Clarke–Wright algorithm [7,89]. Based on the tabu search algorithm, a responsive optimization framework for decision-making in precision agriculture was developed by Kong et al. They presented a mathematical optimization model to generate real-time data. Two metaheuristic algorithms including TS algorithm to achieve decision-support were applied to a hypothetical case study to optimize sugarcane harvesting [92]. Edwards et al. proposed a novel scheduling algorithm working with individual work plans for multiple machines to execute multiple consecutive operations at multiple fields. They utilized two optimization algorithms including standard tabu search [93].

## 5. Benefits of the Agriculture Machinery Movement Optimization

The sugarcane harvesting system in Brazil has been modified mainly in the harvesting and transportation stages. The harvest has transferred from manual to mechanized. Nowadays, the transport of harvested sugarcane is carried out by a tractor and transshipment set which deposits the sugarcane in the truck and transports it to the milling processing center [94].

The study published by Rosa et al. deals with the costs of agricultural production of the sugar-energy sector. This sector has significant representativeness, reaching up to 45% of the total costs of the production chain. The operational costs for harvesting and delivering consist of costs incurred for mechanized harvesting (44%), transshipment (22%) and transportation (33%) [95].

Mechanized sugarcane harvesting, transshipment and truck loading are challenging and complex activities requiring mathematical and computational techniques to assist in decision-making. Díaz and Pérez proposed a computer simulation study aiming at optimization of the process of sugarcane harvest and transportation. The outcome caused better progress and planning in the harvesting process of sugarcane [96].

Sugarcane is grown in rows that can be damaged by undesired machine traffic (across or upon rows). Such damage affects the ratoon from which the crop has to be re-established and re-grown again. It is expected that a number of five to six crop re-establishments from the same ratoon normally takes place until new implantation of the crop. It makes the crop very responsive to controlled traffic farming [97]. Because of this control, the count of machine traces is a multiplication of rows and headlands for machine turns which are excluded, preventing overrunning of the machine.

Due to avoiding the cost of sugarcane mechanization, more and more efforts have been made to redesign the characteristics of the land. This was performed by moving the roads and laying terraces to create the intended rows of crops [98].

It is not enough just to use sugarcane harvesting machines. The harvesting depends on a careful systematization set in the planting area, such as leveling of the land, size of fields, removal of strange materials, lease of roads and carriers, conservation system and furrow planning [99].

Cervi proposed calculating operational and economic performance indicators for sugarcane mechanic harvesting. He obtained data from the case study performed near Botucatu (Brazil) in 2011. After this, Cervi compared operational and economic indicators obtained from the optimized scenario (where a mathematical model for a maximized sugarcane production was used) with indicators obtained from the initial scenario. Linear programming was used to obtain increased production by 17.25%. In addition, due to the raised production, the process throughput of machines carrying out the activities of cutting and loading was improved. This helped to reduce the total cost of slicing and handling by 12% per tonne [100].

Agriculture path planning to minimize the number of turns has been studied by Doriguel and Taix [101,102]. In addition, agriculture path planning to reduce the time for each turn has been studied in [14]. Spekken et al. proposed methods for detecting such tracks in the path planning scope but the standard for profit and power defining the expenditure of an edge for turning around a row row has no proven studies [22,103].

Spekken et al. studied expenses for operation processes for four main field activities with sugarcane: planting, cultivating, spraying and harvesting. Mechanized field activity takes place roughly every five years. A common sugarcane planter pulled by a farm tractor, with a width of 14 m, restricts its steering angles. A tiller connected to a tractor is used to perform three simultaneous operations: cultivation, fertilization and ploughing the soil by chisel. Only the turning capacity of the tractor limits these operations. Sowing and cultivation are limited to two rows of crops whose working width is less than the diameter of rotation. Spraying process is similar to conventional crops due to a width which simply exceeds the rotation limit. Sugarcane is harvested using two transport wagons, which are pulled in a row by a tractor that receives the discs as they move along the combine (comparable to forage harvesting). The combine usually covers one row, but the latest development of the machine has increased the width of the harvest to two rows [22].

Wagons involve the most time-consuming maneuvers due to the substantial steering diameter of the transporter, the lack of headlands of crops and the bounded width of the path. Baio reported that the sugarcane harvester can turn faster than the cane wagon at the ends of rows. Moreover, maneuvers of the cane wagon impair the overall functional field efficacy of the mechanized system. Spraying processes commonly use U-turns and have a run time of 20 s. Navigating space is set among the smallest actual trace width. Omitting rows is not advisable at the time of harvest, as machines cannot miss a row without crossing and damaging one along the harvest line. [104]. Continually, a more detailed maneuver is performed, where the machine moves a specific distance to reach the maneuvering dedicated space (MDS) to steer and go back parallel to the combine. This maneuver is shown in Figure 4. Here, it is intended as a P-turn because the turn resembles the shape of a "P" [22].

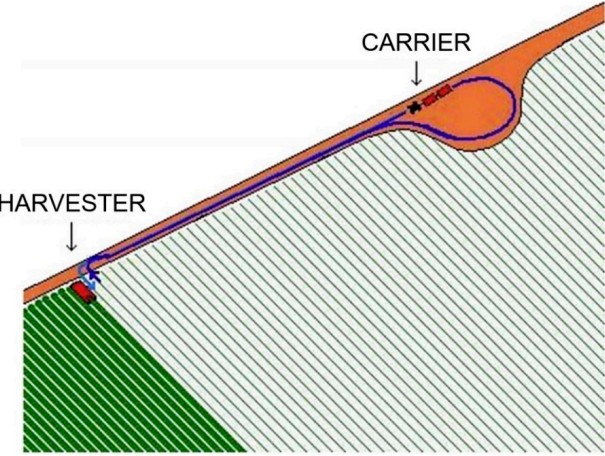

**Figure 4.** Long maneuver of a two-wagon carrier preventing the crop place with overrunning [22].

As can be seen in Figure 5, Spekken et al. modeled the turning maneuvers for the four turns, Ω-turn (a), U-turn (b), T-turn (c) and P-turn (d), in order to establish the time and space for turning required to determine the cost of the maneuver. DBT is the space within steering in a U-turn, DBA is the distance between the implement and front tractor axles in a T-turn and D-MDS represents distance traced by the tractor to reach the maneuver dedicated space before/after leaving the field within a P-turn. Furthermore, the indirect and direct energy inputs demanded by the machine to obtain the energy to time ratio are calculated. The inputs to sugarcane or supplies necessary for its production with regard to the operation of the machine are stated by fuel and labor, machine and hydraulic oil [22].

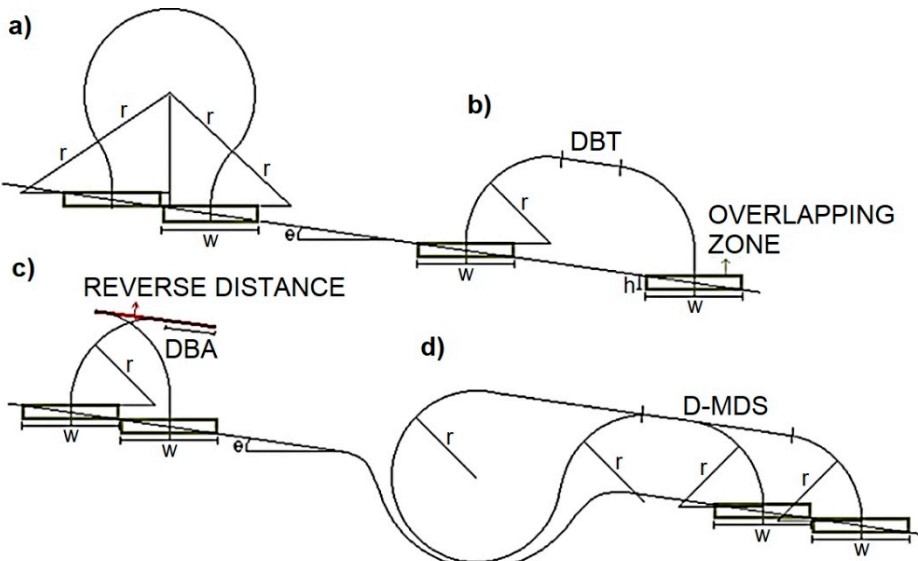

**Figure 5.** Composition of four different types of maneuvers: (**a**) Ω-turn; (**b**) U-turn (DBT is the space within steering); (**c**) T-turn (DBA is the distance between the implement and front tractor axles); (**d**) P-turn (D-MDS is the distance traced by the tractor to reach the maneuver dedicated space before/after leaving the field) [22].

The cost of maneuvering in sugarcane was calculated by a number of variables including machine expenses over time, including energy and efficiency. In a general situation, potential and financial expenses per maneuver amount to 93.3 MJ and $4.45, respectively. Harvesting contains the greater number of the introduced expenses. The price for the maneuvering area depends on the width of the cover and the stretch of the rows. If a group of common numbers is taken into account, it is found that the price of maneuvering is 2073 MJ/ha and 98.9 $/ha [22].

## 6. Discussion

In agriculture, most requirements for crop productivity are related to uncontrollable factors such as rainfall quantity and time, phytopathogenic agents, microorganism attack and insects or aspects related to the environment. In order to make a farm profitable, certain factors such as the optimization of movement of agricultural machinery, precision fertilization and regionally appropriate crop genetics represent fundamental aspects in management for sustainable development.

The main mechanized activities required for most crops of agricultural interest are soil preparation, sowing, mineral fertilization, herbicide and insecticide application (if needed) and harvesting and storage depending on the purpose of the crop. In addition to the operations mentioned, the intense compaction of the soil has to be considered in the case of sugarcane. This operation directly negatively affects the productivity of the crop, since the same planted crop will be harvested for four, five or more years depending on the productivity of the area [105]. This justifies the great interest and efforts of researchers for agriculture machinery movement optimization of this plant in Brazil.

Modern technologies like mathematical analysis using algorithms and an economical approach to mechanical optimization have different management levels tasks for agricultural machinery (Figure 6). This definition is important to show that machinery optimization relies on cooperation in each level to be successful.

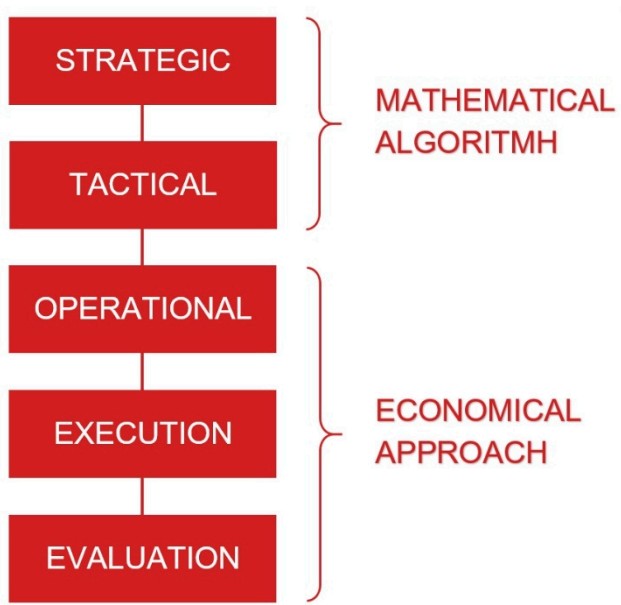

**Figure 6.** Mathematical and economical approaches according to the management level on a farm.

Using the mathematical approach of the algorithm in relation to dealing with the above-mentioned issues is more related to the strategic level of management [3,106]. Meanwhile, the mathematical approach based on economic and energetic use is more closely linked to estimates and data collection on operational and execution levels [22,100].

Strategic planning is an important tool to minimize costs and environmental impacts. Baio developed and tested a linear programming model to assist in selecting a boom sprayer based on the inexpensive hourly costs. It managed the implementation of sensitivity analysis, which was invented shortly after [107]. Klaver et al. developed software for calculating the energy demand of agricultural machinery and tools used to perform field operations (from soil preparation to sowing) [108]. Søgaard and Sørensen created a model system to help with finding the suitable level of field mechanization with regard to technical capability [109].

Tactical scheduling will ensure that planning will be efficient by testing in the field. The advance of navigation systems into commercial agriculture machines has allowed the maneuvers to have a previously planned map of the area (whereas, in the past, the maneuvers were controlled only on the basis of the operator's experience). Sørensen et al. developed an evaluation tool that enveloped the whole string of the manure management system, from the animal shelter to the fields. This tool allows a system-oriented assessment of labor request, device capacity and manure management expenses [110].

To optimize the operational process, Magalhães et al. designed a system allowing synchronization of the combine machine with the carriage. It enables users to avoid crop losses as well as providing operational capacity since one of the problems associated with the harvesting of sugarcane is the absence of a matching mechanism between the combine and the field wagon [111].

Taylor et al. retrieved GPS information from mapping of 23 fields to obtain machine control information. Land capableness and capacity were discovered. Needed time to accumulate, turn and unload the crop were evaluated. Optimizing field models to reduce turns seemed to have greater potential to improve field efficiency than unloading in motion [112]. Hansen et al. investigated trace patterns for a sole combine with specific focus on turns in the headland of the land. The model was

then tested using field data obtained from maize yields. The size of corn head and area of fields had a significant effect on vacant time due to the rotation in the headland [113].

Evaluating the performance of machines is around the last phase of the field operational planning and control round. An important point is the comparative relation between the plotted activity and the currently performed activity. The outline of such a comparison should be incorporated into the next planning cycle and will help the manager to react to the operational projected activity [22].

## 7. Conclusions

Agricultural machinery has been evolving for a long period along with human civilization itself. Desirable optimized processes are the key to sustainable growth. Research has approached this subject using different tools such as metaheuristic algorithms, linear programming, machinery efficiency, costs and energy estimation, data collection and analyses or case studies.

It is important that each case is studied separately. Analysis of the landscape, type of soil, the slope of the area, plant characteristics and machinery operations are needed because, in some cases, optimization can be performed even if the costs are higher compared to the conventional way. Martins evaluated the functional and financial effectiveness of strip deep tillage system and mechanized sowing of sugarcane, where two soil preparation systems were used. The first soil preparation system was conventional, while the second soil preparation system was based on the deep tillage with a subsoiler and rotary spade using a mechanized planting system for both operations. The fuel usage of machinery during operation, the quality of the sowing, the morphological specification of the crop, the functional expenses of the reparation and mechanized sowing processes were assessed. The greatest fuel usage during sowing activity appeared in the standard establishment. Greater operational expenses were discovered with deep tillage in comparison to conventional tillage [114]. On the other hand, a price analysis showed that the profit proportion of the deep tillage was more feasible because of the greater obtained effectiveness. In this case, it was possible to increase productivity and profits using the same amount of land that justifies a higher cost per hectare.

Using programming tools to optimize spatial configuration, route and path planning requires computational time to find a solution that fits the problem. Some solutions can take very low computational time (in the order of milliseconds) [115]. Other solutions tend to substantially increase for bigger areas or require high computational time [116]. Other research such as that of Oksanen and Visala and Palmer et al. offers offline solutions [117,118].

Papers focused on the use of algorithms tended to evaluate the planning phase (including the method of dividing the area and planning the location of the routes and paths), while papers focused on the economic evaluation rather considered the operational and evaluation step of the machinery optimization. Therefore, the algorithm approach on a strategic and tactical level and the economical approach on execution and evaluation levels represent perspective strategies that complement each other. These strategies are able to explain more accurately how to work with not always controllable variables (e.g., fuel and time consumption) [18].

The combination of new technologies, solutions from other fields of study and implementation of autonomous vehicles is a driver of process optimization in agriculture. On the other hand, even though the benefits of new technology are clear, the farmer may not take them into account and may not evaluate them as worthwhile. Using the economic approach can speed up the implementation of a new methodology or technology because discussion about costs and benefits is a language that a lot of farmers with sufficient financial resources and access to compliant infrastructure can understand.

**Author Contributions:** Conceptualization: C.E.B., R.B., P.C., M.F. and P.O.; writing—original draft preparation: P.B., M.F. and P.C.; writing—review and editing: C.E.B., R.B., P.C., A.D., M.F. and P.O.; methodology: P.C.; visualization: T.Z.; supervision: P.B., P.F., P.K. and M.X. All authors have read and agreed to the published version of the manuscript.

**Funding:** This research received no external funding.

**Acknowledgments:** This work was financially supported by the University of South Bohemia, the research project GAJU 059/2019/Z, "Environmentally friendly technologies for sustainable agricultural development".

**Conflicts of Interest:** The authors declare no conflict of interest. The funders had no role in the design of the study; in the collection, analyses, or interpretation of data; in the writing of the manuscript, or in the decision to publish the results.

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
