# Peer review of "Advanced Computational Methods for Agriculture Machinery Movement Optimization with Applications in Sugarcane Production"

_agriculture, doi:10.3390/agriculture10100434_

Round 1

Reviewer 1 Report

An interesting topic was taken up in the manuscript, however, some significant corrections are necessary for the paper to be published.

Detailed comments are included in the file.

Author Response

Dear Reviewer,

thank you for your comments. Based on your observations, we made following changes of our manuscript:

1) According to the opponents' recommendations, we have clearly defined the professional focus of the article and its goals. We achieved this by editing the relevant text sections and changing the title of the article, which is "Advanced Computational Methods for Agriculture Machinery Movement Optimization with Applications in Sugarcane Production."
2) We have expanded the article with a separate chapter, which deals with the methodology according to which we performed an analysis of literary sources.
3) According to the recommendation of the second opponent, we have included in the report several other professional works published by other authors on the issue in the last few years. These works are classified according to the used calculation method and now form a separate chapter of the article.
4) We have narrowed the overview of the benefits that the application of calculation methods brings to the area of sugar cane cultivation. This is a key crop in Brazil, the home country of one of the authors of the article under review.

Sincerely

Martin Filip

Reviewer 2 Report

  1. General conclusions:
  2. Congratulations for the paper!
  3. Please, try to compare the title, aims in abstract (L20-22:“The aim of this paper is to review modern approaches to agriculture machinery movement optimization and to evaluate the economic benefits achieved by their application in production processes”), and aim presented in Introduction (L57-59:“This review aims to discuss modern approaches in agriculture optimization machinery using mathematical models such as heuristic methods or mathematics analyses as well as economic methods based on time and equipment efficiency.”).
  4. Based on previous, I consider it is necessary to insist more on the economic benefits, also in Conclusions.

  1. Particularly observations
  • Please, explain: how “in-field nonworking distance”, and “total non-working distance” (L118, 119), can be compared with “Total distance” (L113), for evaluate the methods and economic benefits?
  • L123-129: Could be concluded there are less differences (less 1,4%) between both methods, but with specific observations.
  • L156-157: “The operational costs of field production consist of costs incurred for mechanized harvesting (44%), transshipment (22%) and transportation (33%)”, - I think must be operational costs for harvesting and delivering, not all operational costs.
  • L328-330: I think the final conclusion: “Using the economic approach can speed up spreading a new methodology or technology because discussion about costs and benefits is a language that everybody can understand”, could be improved if the text stop at verb “can” (some farmers cannot access the financial resources, or the infrastructure are yet missing etc.).

Author Response

Dear Reviewer,

we thank for your comments. Based on your comments, we made following changes:

-Additional descriptions of the terms "total distance" and "non-working total distance" have been added to the text for better understanding.

-Proposed observation regarding comparison between results (lines 288, 289 in revised manuscript) has been accepted and integrated to the text.

-By analyzing the source publication, we have come to the conclusion that presented values are indeed related to harvesting and delivering costs. This observation was taken into account when editing the text (line 383).

-The last sentences of the conclusion have been modified to also take into account farmers with limited resources.

Detailed description of performed changes of our manuscript is presented in attached coverletter.

Sincerely

Martin Filip

Round 2

Reviewer 1 Report

The Authors addressed four of the previous five comments. Most of the shortcomings in the paper were corrected. Unfortunately, the third comment was not considered in the new version of the manuscript.

Said version contains a sentence (line 65-67):

In the second and third section we reviewed recent mathematical models and computational methods based on artificial intelligence which are generally used in agriculture.”

However, in the second part (2. Materials and methods) no such thing was included.

In the third part (3. Agricultural routing problem (ARP)) ARP optimization is defined and similarly  no overview of the methods is provided.

On the other hand, in the fourth part (4. Metaheuristic algorithms for agricultural applications), the Authors significantly characterized several optimization methods, for which they found publications regarding their use in sugarcane production. Not all of these methods belong to artificial intelligence methods, and they are not all of optimization methods used in agriculture. Already in the previous review, I suggested that the Authors refer to publications that would allow them to systematize and correctly classify optimization methods. Unfortunately, this has not been done. It is necessary to correct this fragment of the paper.

The first method that does not require a significant amount of work is to remove statements from the manuscript that suggest that the publication has carried out a comprehensive analysis of optimization methods using computational artificial intelligence tools in agriculture. They should be replaced with the statement that the optimization methods described were those for which applications were found in sugarcane production (according to the publication search methodology used).

The second, much more labor-intensive way is to actually conduct a review of the latest mathematical models and computational methods based on artificial intelligence. In this case, it would be advisable to provide a classification of methods (e.g. using the publications I gave in the previous review) and then provide a short description and examples of agricultural applications.

I have two more detailed comments:

Specific comment 1:

Since the corrected title of the paper is:

Advanced Computational Methods for Agriculture Machinery Movement Optimization with Applications in Sugarcane Production

and corresponds to the content of the work, in my opinion the description of paper’s aim should be modified, which currently is:

line 20: The aim of this paper is to review modern approaches to agriculture machinery movement optimization.

line 64: This review aims to discuss modern computational approaches which take place in agriculture machinery movement optimization.

In Abstract and in Introduction, the aim of the work should be:

(…) modern approaches to agriculture machinery movement optimization with applications in sugarcane production

(…) modern computational approaches which take place in agriculture machinery movement optimization with applications in sugarcane production

Specific comment 2:

I suggest to delete (line 146-147):

“Biologically inspired genetic algorithms like neural networks represent a new computational model having its roots in evolutionary sciences.”

Author Response

Dear Reviewer,

Thank you for your comments. Based on your comment, we have edited the description of individual parts of the text to correspond to the numbering of the chapters. This discrepancy arose from extensive revisions of the text following the previous revision. Furthermore, based on your recommendation, we deleted the first sentence describing the Genetic Algorithm, which was indeed incorrect. In addition, based on your recommendations, we've edited the aim of the article in the Abstract and Introduction. In connection with this, we have also modified the abstract - now it is obvious that in our article we deal mainly with metaheristic methods and not with artificial intelligence in general.

In the new version of the manuscript, we have therefore included your recommendation regarding the absence of a classification of computational methods. Figure 2 provides a comprehensive overview of important metaheristic methods, including methods relevant to agriculture and described in detail below in the text.